



# The experimental characterisation of dynamic stall of the FFA-W3-211 wind turbine airfoil

Simone Chellini[1], Delphine De Tavernier[1], and Dominic von Terzi[1]

[1]Faculty of Aerospace Engineering, Delft University of Technology, 2629HS Delft, The Netherlands

**Correspondence:** Simone Chellini (s.chellini@tudelft.nl)

**Abstract.** In this work, an experimental campaign was carried out to determine for the first time both the static and dynamic aerodynamic properties of the FFA-W3-211 airfoil. This airfoil is widely used in the wind energy community as part of IEA reference wind turbine designs but is lacking experimental data for design, simulation tool validation and dynamic stall modelling purposes. The airfoil model was designed and manufactured to be tested in the Low Speed, Low Turbulence wind tunnel of the TU Delft. The airfoil was tested statically for Reynolds numbers from $Re_c = 5 \times 10^5$ to $Re_c = 3.5 \times 10^6$ and dynamically for up to $Re_c = 2 \times 10^6$, covering the steady, unsteady and highly unsteady aerodynamic behaviour. Data were acquired through pressure measurements at the surface of the airfoil and in the wake, as well as by the use of thermal cameras. The static results highlight a strong dependence of the lift and drag polars on the Reynolds number and indicate the presence of laminar separation bubbles for the lowest static Reynolds number regimes. Therefore, two distinct regimes can be identified for the static data between which a fundamental change in flow behaviour is observed. The dynamic behaviour was studied for the positive, negative and linear regions of the polar. The positive region is governed by the lack of a leading-edge vortex. This is in contrast to the negative region of the polars where the effects of a vortex close to the leading edge dominate. The sensitivity of results to reduced frequency, amplitude, and Reynolds number is discussed.

## 1 Introduction

Wind turbines have seen a tremendous increase in rotor diameters, as testified in the latest available reference wind turbine (RWT) of the International Energy Agency (IEA), the *IEA 22MW RWT*, which features a rotor diameter of $D = 284m$ and a hub height of $h = 170m$ (Zahle et al., 2024). Such a large wind turbine is prone to experiencing significant time-varying inflow conditions along the blades due, among others, to wind shear, turbulence, aeroelastic deflections and yaw misalignment. The force enhancement due to dynamic oscillations is responsible for design loading conditions that are often difficult to quantify and affect the blade's lifespan and overall turbine performance. This is the reason why airfoils that undergo dynamic stall effects have generated renewed research interest. The *IEA 22MW RWT* and its predecessor, the *IEA 15MW RWT* (Gaertner et al., 2020), employ airfoils from the *FFA* family at the outboard sections of their blades with different thicknesses depending on their span location. For a given wind condition, dynamic stall is assumed to be one of the largest sources of load uncertainty for these wind turbines.



Dynamic stall has long been observed as design driving in applications and, therefore, has been often discussed in the literature, see Ham (1967), Leishman (2002), Gardner et al. (2023) and the citations therein to name only a few. While it is important to bear in mind that the mechanisms of dynamic stall vary depending on the airfoil profile and flow conditions, a generalisation of the phenomenon could be described as in the following: The boundary layer separation on the airfoil is delayed above the equivalent static stall angle, after which a *dynamic stall vortex* (DSV) is shed from the airfoil's suction side. This process causes a temporary increase in lift exceeding the maximum steady-state equivalent. A nose-down pitching moment and boundary layer reattachment complete the cycle that forms a dynamic stall hysteresis loop. The loop varies greatly in magnitude depending on the oscillating conditions.

Many experiments have investigated different aspects of dynamic stall and add-ons. In De Tavernier et al. (2021), vortex generators (VGs) effects have been demonstrated in unsteady conditions for the DU17DBD25, an airfoil specifically designed for a sub-MegaWatt vertical-axis wind turbine. The results were acquired for a chord-based Reynolds number of $Re = 10^6$, showing that wind turbine blades would benefit from passive VGs installations for flow control. In Rainbird et al. (2015), the effects of virtual camber have been investigated experimentally for the symmetric NACA0018 airfoil. The experimental campaign was carried out in the Parkinson blockage tolerant tunnel at Imperial College for $Re = 3 \times 10^5$. The polars were compared with a CFD study to highlight that consideration of the airfoil's curvature should be included when comparing results from BEM codes with CFD. The NACA 23012 was tested dynamically at $Re = 1.5 \times 10^6$ in Leishman (1990). The experiment was carried out in the Handley-Page low-speed wind tunnel at the University of Glasgow, where data for the $c = 0.55m$ wing was acquired via 30 pressure taps, highlighting the increasing stall abruptness with increasing Reynolds numbers and the characteristic lift increase associated with the nose-up motion.

Experimental characterization of the dynamic stall phenomena can be challenging, particularly when combining medium-to-high Reynolds numbers with unsteady reduced frequencies. In the work of Carta (1974) and McAlister et al. (1978), dynamic stall was analysed for an airfoil pitching sinusoidally. While simplified, the sinusoidal case provides a good testing benchmark replicating helicopter or wind turbine blades. Patterson and Lorber (1990) compared the results of a medium-level Reynolds number experiment carried out for $2 \times 10^6 < Re < 4 \times 10^6$ for a supercritical airfoil, the $Sikorsky\ SSC - A09$ with two computational methods. The review in McCroskey (1982) provides background for airfoils undergoing transonic flow conditions dynamically.

The DNW-HDG facility is used in Llorente et al. (2014) to compare the aerodynamic performance of two wind turbine dedicated airfoils. Two wind tunnel experiments were carried out: a high Reynolds test in the DNW-HDG and a medium Reynolds in the TU Delft. The campaigns allowed testing across a wide range of Reynolds numbers, from 3 million up to 12 million, with varying turbulence intensity. The airfoils tested were the DU 08-W-180, the DU 91-W2-250 and the AWA18-1. In both experiments, data was acquired through pressure taps distributed on the airfoil surface. The airfoils were extensively tested for static angles of attack, highlighting the higher lift obtained for higher Reynolds for the same angles of attack as well as static stall delay. Results from the pressurized and non-pressurized facilities were compared for the same airfoil and Reynolds number, 3 million. Importantly, the Mach number associated with the two facilities varies significantly for the same Reynolds: this is found to be 0.03 for the Gottingen HDG and 0.23 for the Delft LTT. Despite the geometry and compressibility differences,





results show a good agreement for the static polars obtained in these two facilities. In Ceyhan et al. (2016), measurements on a DU 00-W-212 airfoil are presented, which have been taken in the pressurized DNW-HDG wind tunnel up to a Reynolds number of 15 Million. The high-Reynolds results show a flattening in the aerodynamic efficiency of the wing profile. Recently, dynamic stall was studied for a NACA0021 airfoil in Kiefer et al. (2022) for high Reynolds numbers. The experiment was conducted in a pressurised facility where high Reynolds numbers close to the real-life condition can be reproduced. Thanks to a low free-stream velocity, the facility allows the pure effects of the Reynolds number to be isolated, with no transonic effects due to the low Mach number. Overall, combining high Reynolds numbers with dynamic stall testing has been proven to be difficult. Literature has repeatedly shown the need for a deeper understanding of Reynolds numbers effects for wind turbine airfoils, with a special focus for the more complex transient case, which heavily impacts the wind turbine design (Kiefer et al., 2022).

Dynamic stall has been thoroughly investigated via simulations in the past, often trading off the Reynolds number and reduced frequency. The work of Boye and Xie (2022) investigates the effects of high reduced-frequencies for a NACA0012 airfoil, albeit for chord-based Reynolds numbers of only $Re = 135.000$. More studies were carried out to analyse the pitching of the NACA0012, all for low Reynolds numbers in Tuncer et al. (1990), Choudhuri and Knight (1996), Gharali and Johnson (2013). While the airfoil is often considered a wind turbine benchmark, the low thickness and lack of a cambered profile make it unsuitable for wind turbines. Gardner et al. (2023) recently highlighted the renewed interest in dynamic stall, reviewing relevant work in the field.

Dynamic stall predictions rely on simulation data models, such as the Leishman and Beddoes (1989) model that are not developed specifically for wind turbine tip airfoils characterised by a thickness of $\approx 20\%$ Boutet et al. (2020) and high camber. The importance of the Beddoes-Leishman (BL) model and its correct implementation is shown in Melani et al. (2024), highlighting the experimental data's role in calibrating dynamic stall models correctly. Initial results from Chellini et al. (2024) show the consistent failure of the Beddoes-Leishman model in capturing the transient aerodynamic behaviour, particularly at negative angles of attack, where most dynamic stall models lack experimental calibrations. For instance, despite being based on symmetric airfoil data, the BL model fails to replicate the experimental separation and load enhancement associated with negative angles of attack. In wind turbines, the model is widely adopted in popular tools such as OpenFAST, leading to inaccurate load reconstructions at negative regimes (Chellini et al., 2024).

To understand the physical features of dynamic stall on relevant wind turbine airfoils, this work presents the results of the *FFA-W3-211* airfoil tested in static and dynamic conditions. The experiment is carried out for a range of static and dynamic inflow angles and steady to highly unsteady reduced frequencies, aiming to characterise three distinct flow regions in the negative, linear and positive regions of the polars.

## 1.1 Objective and paper structure

This work presents the first experimental data for the FFA-W3-211 airfoil, quantifying the static and dynamic aerodynamic behaviour. The main objective is to determine how the combination of reduced frequency and Reynolds number affects the





airfoil's loading. The analysis is carried out to understand the driving factors affecting the flow physics around the airfoil. Firstly, the corrections procedure is introduced. This is followed by validating the newly acquired data against existing experimental results in §3.1. The analysis of the static and dynamic loads follows in §3.2. The analysis is divided into Reynolds number, reduced frequency, oscillating frequency and amplitude effects, isolating all the experimental and relevant flow physics parameters.

## 2 Experimental set-up

### 2.1 Wind Tunnel Facility

The experimental campaign is carried out in the *Low-Speed Low-Turbulence wind tunnel* (LTT) of Delft University of Technology, a closed-section recirculating wind tunnel. The facility features changeable octagonal testing sections, with a width of 1.80m, length of 2.60m and height of 1.25m. The octagonal testing section returns a cross-sectional area of $A_{ltt} = 2.07m^2$. Each testing section is equipped with a mechanically actuated turntable flush to the tunnel's top and bottom walls, which allows the airfoil with a precision of $\pm 0.1°$. The turbulence level varies from $0.015\%$ at $20m/s$ to $0.07\%$ at $75m/s$ due to the large contraction ratio of 17.83. The six-bladed fan is powered by a 525 kW DC motor, returning a free-stream velocity up to $U_\infty = 120ms^{-1}$, and Reynolds numbers up to $Re_c = 3.5 \times 10^6$ based on the airfoil's chord can be tested. More details can be found in Delft (2017). The full diagram of the facility can be found in Figure 1.

### 2.2 Airfoil model

An FFA-W3-211 airfoil model with a maximum thickness of $21.1\%$ chord (Fig. 2) is manufactured from carbon fibre composite. The airfoil model has a chord length of $c = 600$ mm and a length of $h = 1246$ mm to span the LTT test-section height, resulting in an aspect ratio $AR \approx 2$, in line with previous airfoils tested in this facility (e.g., Baldacchino et al., 2018; Timmer and Van Rooij, 2003; Llorente et al., 2014). The trailing edge thickness is measured as $1.7$ mm. The airfoil is connected vertically to the wind tunnel's walls through an aluminium shaft, which exits the wind tunnel on the upper side to be connected to a linear actuator. A metal insert secures the shaft to two outer wooden ribs. The airfoil is fixed by two ball bearings mounted on support plates and two safety pins for a total of four contact points. The plates are inserted inside the wind tunnel turntable rails and secured flush to the testing section.

A total of 93 pressure taps, refined around the leading edge, with a diameter of 0.4 mm, are distributed on the airfoil skin. The taps distribution is angled at $\approx 15°$ to reduce any potential boundary layer disruption between the orifices. Each pressure tap is connected to an $\approx 2.7$ m long tube, which exits the airfoil model through a lower aperture located at around $x/c = 0.5$. Importantly, this represents a limitation for the airfoil's oscillating amplitude, which can operate safely between $-25° < \alpha < 25°$. The airfoil model was polished to a mirror-finish texture, and the flow transition from laminar to turbulent was verified with thermal cameras to ensure that no model imperfection provoked early transition, as shown in Figure 3, and prove the 2D uniformity of the model. A comparison between $Re_c = 1.0 \times 10^6$ and $Re_c = 3.5 \times 10^6$ is visible in figure 3.





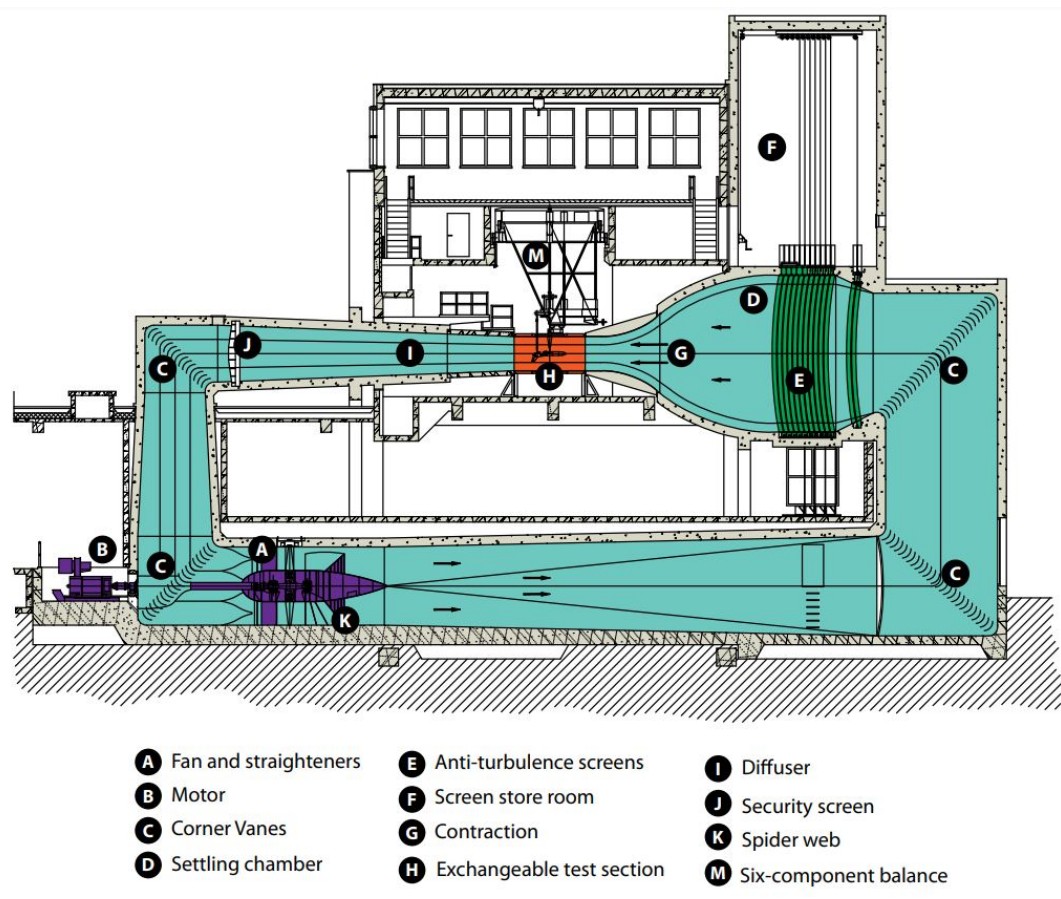

| | | | | | |
|---|---|---|---|---|---|
| **A** | Fan and straighteners | **E** | Anti-turbulence screens | **I** | Diffuser |
| **B** | Motor | **F** | Screen store room | **J** | Security screen |
| **C** | Corner Vanes | **G** | Contraction | **K** | Spider web |
| **D** | Settling chamber | **H** | Exchangeable test section | **M** | Six-component balance |

**Figure 1.** Diagram of the low-speed, low-turbulence wind tunnel at TU Delft. The flow reaches the testing section from right to left.

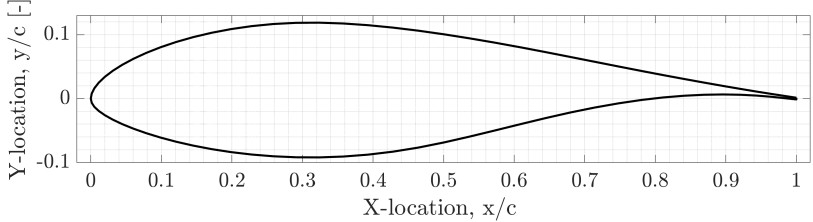

**Figure 2.** Cross section of the FFA-W3-211 airfoil. The cambered airfoil has a thickness of 21.1 %.





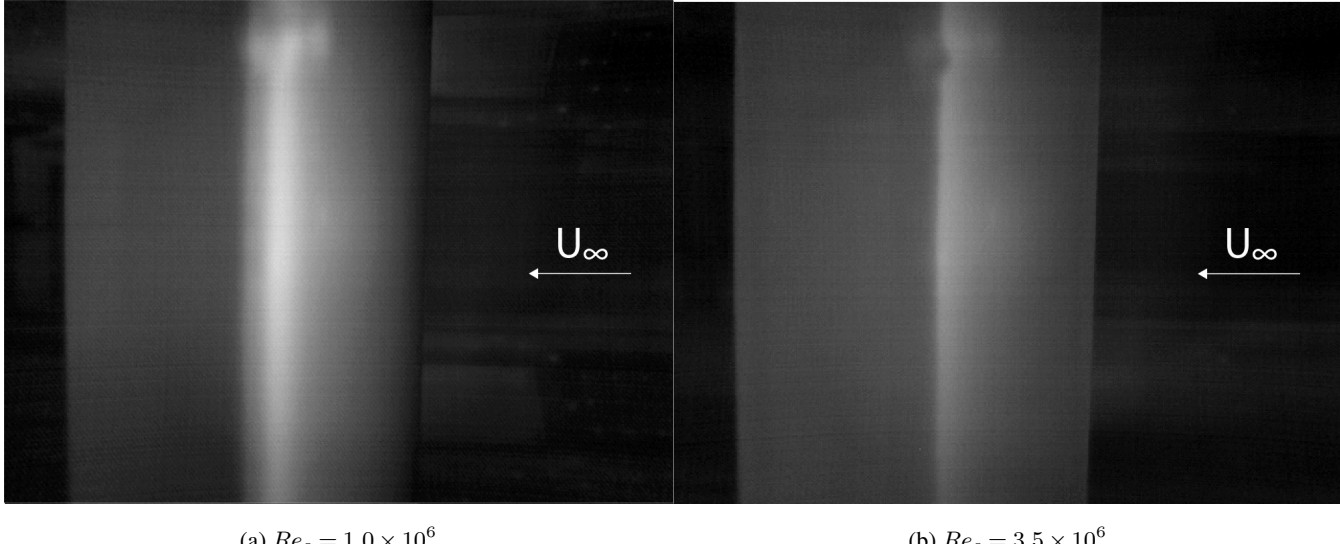

(a) $Re_c = 1.0 \times 10^6$          (b) $Re_c = 3.5 \times 10^6$

**Figure 3.** Qualitative transition location on the airfoil's suction side for $\alpha = 0°$.

### 2.3 Data acquisition

#### 2.3.1 Static tests

The 2D flow data was acquired through the 93 pressure taps, connected to the pressure scanner through four adapters. The pressure measurements at the leading and trailing edges were split into two channels for redundancy. The pressure scanner was also used to acquire the total and static pressure in the wind tunnel. In static cases, a wake rake is used to quantify the loss of momentum in the wake. This comprises 67 pressure tubes and 16 static tubes over a length of 504 mm. The pressure rake automatically moves in the horizontal direction to centre the wake and traverses along the spanwise direction for each acquisition to average potential three-dimensional flow effects. When the rake can no longer capture the airfoil's turbulent wake, pressure drag is used instead. Data is sampled at a frequency of $5Hz$. Two $2mm$ gaps are found between the wind tunnel and the wing on each side. The gaps were sealed with tape for the static measurements, minimising the loss of momentum from the lower aperture, from which the pressure tubes exit the model.

The airfoil is rotated through the turntable for the static polars acquisition around its quarter-chord point. The static polars are acquired for angles of attack of $-20° < \alpha < 20°$. Local refinements are carried out to characterise the stall region and the design lift-to-drag ratio. The static experiment is carried out for $5 \times 10^5 < Re_c < 3.5 \times 10^6$. Therefore, the equivalent Mach number for the highest Reynolds number case is evaluated as $M = 0.25$, and compressibility effects are neglected in this analysis. Depending on the boundary layer thickness, Reynolds number and angle of attack, impurities in the boundary layer can disrupt the pressure readings from the airfoil's tap, especially towards the leading edge, voiding part of the dataset. Therefore, the airfoil model was regularly cleaned to ensure a smooth, dust-free surface and prevent any undesired boundary layer transi-



tion.

Following the clean airfoil tests, a zigzag tape was applied to the airfoil at $x/c = 5\%$ on both the suction and pressure sides. Different tape thicknesses were considered to ensure a complete boundary layer transition to turbulence independent of the Reynolds number and angle of attack. Three different zigzag tapes were tested, with a thickness of $0.4, 0.5, 0.65mm$ to transition the flow at $x/c = 5\%$ on the pressure and suction side. Transition to turbulence was observed for Reynolds numbers in the range $5 \times 10^5 < Re_c < 3 \times 10^6$ and for multiple angles of attack. It was decided to employ a $0.4mm$ zigzag tape to trigger

transition for all cases with a reduced drag penalty compared to thicker zigzag tapes. The lower and upper-pressure distributions obtained from the pressure taps on the airfoil's surface are integrated to determine the normal coefficient (Eq. (1)), where the prime indicates the uncorrected coefficients.

$$C_n' = \int_0^1 (C_{p,l} - C_{p,u}) \, d\left(\frac{x}{c}\right) \tag{1}$$

Where $C_{p,l}$ and $C_{p,u}$ are the lower and upper-pressure distributions, respectively. The drag coefficient is obtained through the

wake rake as Eq. (2)

$$C_d' = \frac{2}{c} \int_{\text{wake}} \sqrt{C_{p,t} - C_{p,s}} \left(1 - \sqrt{C_{p,t}}\right) dy \tag{2}$$

The normal and drag coefficients are used to quantify the 2D lift in Eq. (3).

$$C_n' = C_l' \cos(\alpha) + C_d \sin(\alpha) \quad C_l' = \frac{C_n'}{\cos(\alpha)} - C_d' \tan(\alpha) \tag{3}$$

### 2.3.2 Dynamic tests

The wing model's shaft was connected to a UniMotion PNCE 40 BS 1610 linear actuator for dynamic testing purposes. The actuator's extrusion is connected to the airfoil's steel shaft to couple the linear and pitching motions, and its extrusion is controlled through a LabVIEW script. An angular encoder is coupled to the airfoil's shaft, providing the real-time angle of attack reading. More details regarding the setup can be found in De Tavernier et al. (2021). Data is sampled from the wing's pressure taps with the same methodology described in the previous section, with a frequency of $300Hz$. The dynamic polars

were acquired by pitching the airfoil around its quarter chord point.

Depending on the flow conditions, different actuator calibrations were needed. The increased aerodynamic loads for the clean airfoil configuration shifted the zero angle of attack for the equivalent transitioned flow case. Additionally, relevant calibration differences were observed after the static stall angle and as a consequence of increasing Reynolds numbers. As a representative example, the initial arm position reported a half a millimetre difference between transitioned $Re_c = 2 \times 10^6$ and clean $Re_c =$

$1 \times 10^6$. This may not seem like a lot; however, translated into an effective angle of attack, this difference is responsible for an offset of $\pm 0.3°$ for the pitching motion.



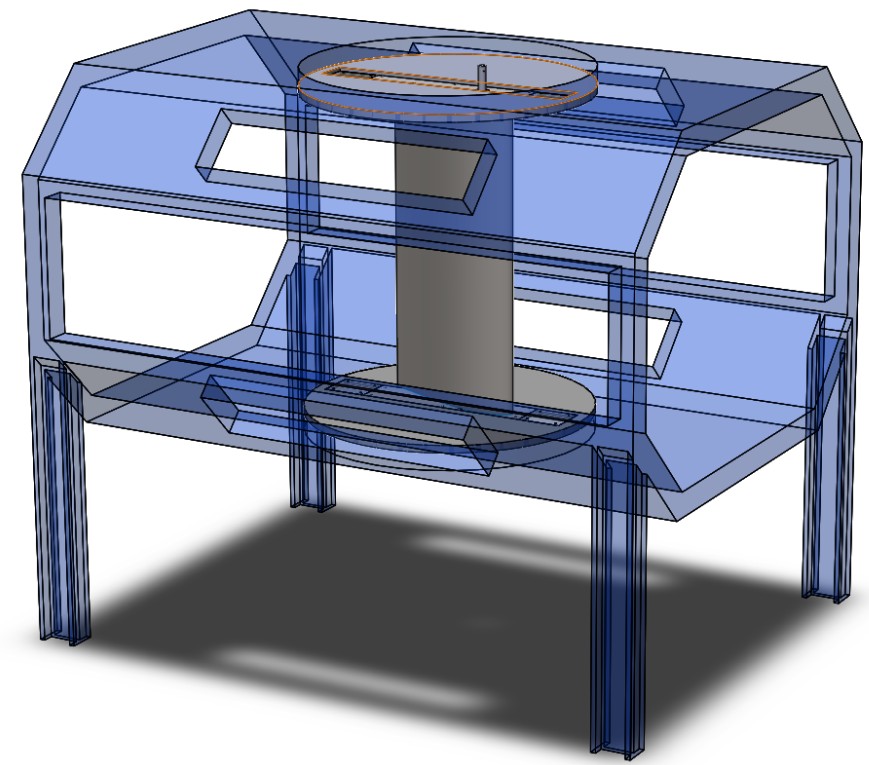

**Figure 4.** CAD diagram of the wind tunnel section. The turntable walls hold the airfoil in place and rotate for static data acquisition.

A representative, yet simplified, oscillation case for a wind turbine airfoil is defined by a simple sinusoidal motion, described in Eq. (4) as

$$\alpha(t) = \alpha_0 + A sin(2\pi f t) \tag{4}$$

Where $\alpha_0$ is the initial mean angle of attack, $A$ is the oscillating amplitude, and $f$ is the oscillating frequency in Hertz. The actuator was controlled through a LabVIEW code, setting the mean angle of attack, oscillating amplitude, frequency and acquisition time. The actuator is secured above the wind tunnel testing section and bolted into the turntable panels.

The frequency value from Eq. (4) also affects the reduced frequency, which is defined here as Eq. (5)

$$k = \frac{\pi c f}{U_\infty} \tag{5}$$

Where c is the airfoil's chord and $U_\infty$ is the free-stream velocity. This non-dimensional parameter describes the unsteadiness of oscillation. Three separate unsteady regimes are identified in Corke and Thomas (2015). The flow field can be considered unsteady for $0.05 \leq k \leq 0.2$, and for $k \geq 0.2$, highly unsteady. Below $0.05$, the unsteady effects can be neglected. To capture the three different flow regimes, a comprehensive span of Reynolds numbers and reduced frequency is acquired for the negative





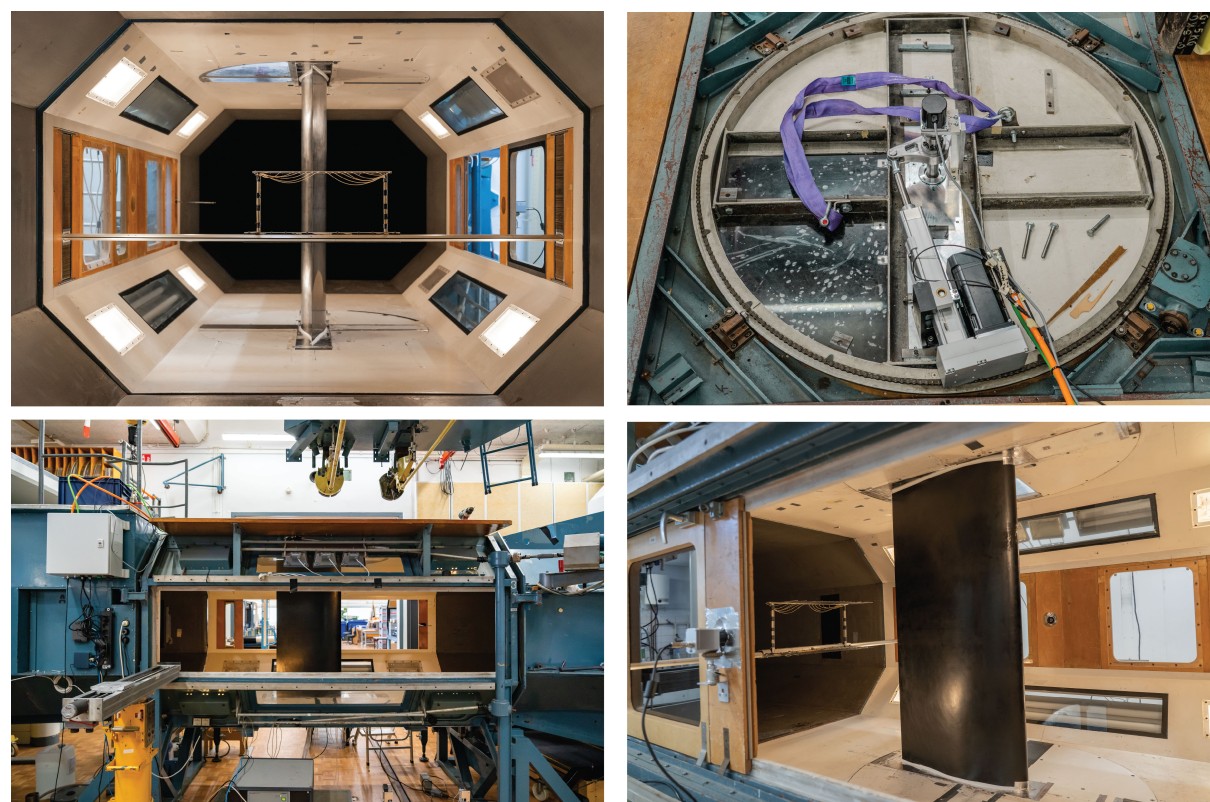

**Figure 5.** Experimental setup details. The top left image shows the testing section geometry, with the wake rake behind the taped airfoil. The linear actuator mechanism is shown in the top right image. The testing section installed in the wind tunnel is shown in the bottom left image, without the covering panels on. The airfoil model taped to the wind tunnel walls is shown in the bottom right image, with the thermal cameras installed at the airfoil's midspan, for $x/c = 50\%$

and positive stall regions and the linear region of the polars. The dynamic matrix is repeated for the clean case and for the transitioned case, using the same zigzag tape described in the previous paragraph.

Together with the local pressure on the airfoil, the atmospheric and operational conditions were recorded through the wind tunnel measurement devices, highlighting a variation in the instantaneous free-stream readings due to the changing blockage in the wind tunnel. Data was acquired starting from the negative mean angle of attack, first acquiring the lowest frequency and amplitude for 50 loops of dynamic stall.

Finally, low-frequency cases were acquired with the same methodology to determine the reduced frequency at which the dynamic effects disappear for the polars, for which data was acquired at $30Hz$. The experimental matrix is found in Table 1, where one parameter at the time was changed.







**Table 1.** Dynamic experimental matrix. Each parameter is changed independently of the others, returning four independent degrees of freedom in the experimental matrix.

| Reynolds number [/] | Frequency [Hz] | Amplitude [°] | Mean Angle [°] |
|---|---|---|---|
| $Re_c = 5 \times 10^5$ | 0.6 | $\pm 3$ | $-9.4$ |
| $Re_c = 1 \times 10^6$ | 1.2 | $\pm 5$ | $-1.4$ |
| $Re_c = 2 \times 10^6$ | 1.8 | $\pm 7$ | 9.6 |
| / | 2.4 | $\pm 9$ | / |
| / | / | $\pm 11$ | / |

## 2.4 Measurement Corrections

### 2.4.1 Static data corrections

The static raw pressure data is corrected using the wall effects corrections found in Dalton (1971), correcting for lift interference, wake blockage, solid blockage, and streamline curvature. Here, the uncorrected drag is used to correct for wake blockage, directly affecting the corrected lift coefficient. At high angles of attack, the turbulent airfoil's wake becomes too wide to be captured by the rake. Depending on the Reynolds number, the wake is moved outside the wake. Therefore, the drag is determined by integrating the pressure distribution along the airfoil surface, which is common practice (De Tavernier et al., 2021).

The wake blockage is defined in Eq. (6) as

$$w_b = C_D \frac{0.25h}{c} \frac{1 + 0.4M^2}{\beta^2} \tag{6}$$

Where $M$ is the Mach number, and the compressibility factor $\beta$ is defined as $\beta = \sqrt{1 - M'^2}$. The wind tunnel blockage factor, $\sigma$, compressibility factor $\beta$ are defined in Eq. (7)

$$\sigma = \frac{\pi^2}{48} \left( \frac{c}{h} \right)^2 \tag{7}$$

The angle of attack is corrected using Eq. (8)

$$\alpha = \alpha' + \frac{57.3\sigma}{2\pi\beta} (C'_l + 4C'_m) \tag{8}$$

### 2.4.2 Dynamic data corrections

Due to the dynamic oscillations, retrieving the drag coefficient readings from the wake rake was impossible. This was due to the large and variable wake shed by the airfoil, which the rake would not fully capture. The drag coefficient is used in the static





corrections to obtain the lift coefficient, and it is not possible to transition from normal and tangential coefficients to the lift coefficient without the drag quantification.

An initial delay and amplitude loss are found between the pressure sampled at the skin of the airfoil and the DAQ values. Considering the pressure tube length, width, and testing conditions, the pressure lag and attenuation can be evaluated in the post-processing using the corrections from Bergh and Tijdeman (1965). The method has been coded in the software tool

PreMeSys V2.0, validated in De Tavernier et al. (2021) for dynamic airfoil testing. The corrected pressure values are then translated into instantaneous forces on the airfoil by integration.

While a methodology concerning aerodynamic static corrections has been validated, evidence of dynamic corrections has yet to be found in the literature. Previous literature such as Kiefer et al. (2022) and De Tavernier et al. (2021) display the dy-

namic polars results uncorrected for any of the traditional voices mentioned in the last paragraph. A preliminary study assessed the impact of drag coefficient in the static corrections and validated the corrections from Dalton (1971) for the pitching airfoil case. Figure 6a shows the differences between the corrected, uncorrected and partially corrected results. The lift-curve slope is shown for $Re_c = 10^6$, with corrections from Allen and Vincenti (Dalton, 1971) fully applied, partially applied and uncorrected. There is good agreement between the fully and partially corrected data regarding the location and value of $C_{L_{max}}$ and pre-stall

slope. However, after the stall angle is reached, the effects of wake blockage become predominant in the polars corrections, with the partially corrected data diverging from the fully corrected data as the angle of attack increases and accounting for almost $50\%$ at $\alpha = 20°$.

The safety pins that block the airfoil to the wind tunnel testing sections were removed to pitch the airfoil around its quarter chord

point. Thus, the tape that sealed the 2mm gap between the airfoil walls and the wind tunnel was removed, exposing the gap on the lower plate that allows the pressure tubes to exit the model. At low angles of attack, the airfoil cross-section fully or partially covers the gap, limiting the loss of momentum. However, as the angle of attack increases, the gap is more exposed to the free-stream flow. A calibration was carried out ahead of the dynamic acquisitions to assess the loss of momentum through the wind tunnel aperture. The effect of the sealing tape was quantified for three static cases, for $Re_c = 5 \times 10^5, 1.5 \times 10^6, 2.5 \times 10^6$, as

shown in figure 6b. The linear region slope is reported for $-6° < \alpha < 6°$ in Table 2. The results show a good agreement in the linear region of the polars for all cases, with the difference in the pre-stall slope decreasing with increasing Reynolds numbers. This is due to the higher flow inertia that, for higher free-stream velocities, allows the suction of the wind tunnel cavity to be overcome. As the static stall angles are approached, the loss of momentum becomes more evident for low Reynolds numbers while still showing similar stall trends across all cases.

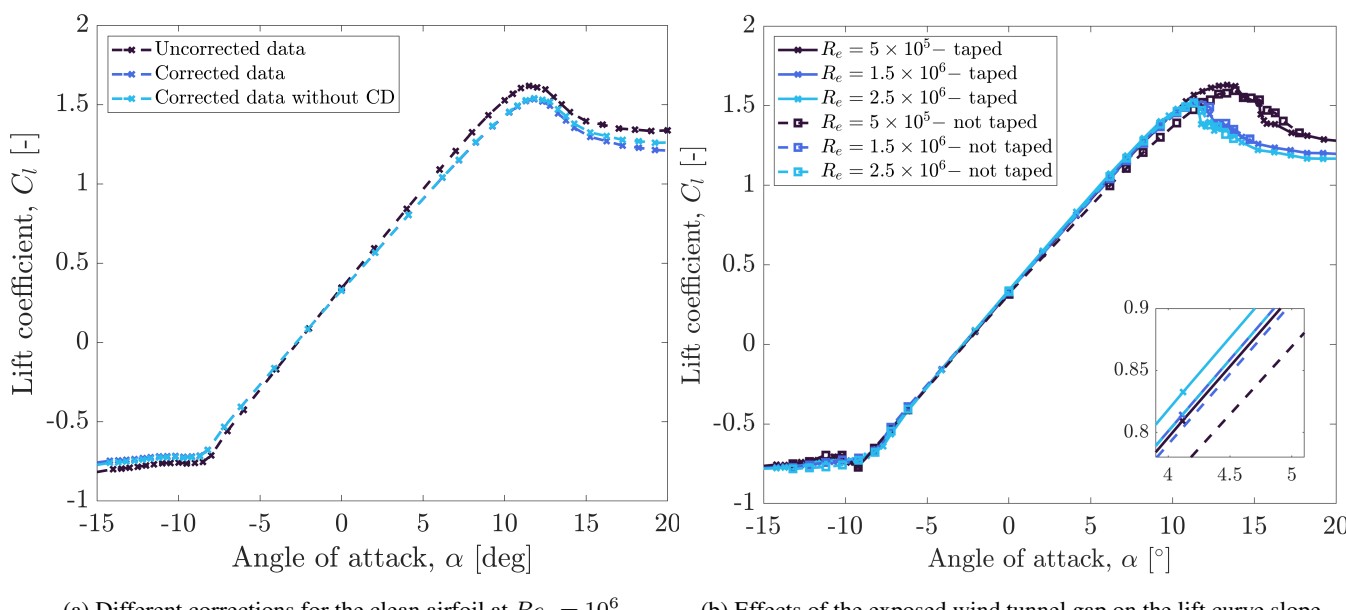

(a) Different corrections for the clean airfoil at $Re_c = 10^6$.

(b) Effects of the exposed wind tunnel gap on the lift curve slope.

**Figure 6.** Effects of the wake blockage term and sealed wind tunnel gap in the static lift coefficient curve

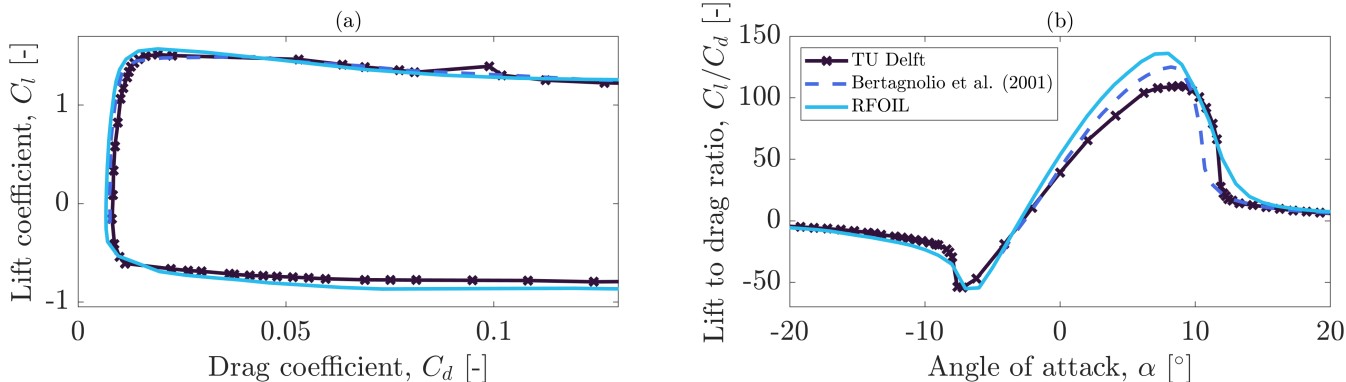

**Figure 7.** Clean polars comparison for the FFA-W3-211 airfoil for $Re_c = 1.8 \times 10^6$





**Table 2.** Lift curve slope calibration in the linear region of the polars.

|  | Sealed gap slope | Open gap slope | Difference |
|---|---|---|---|
| $Re_c = 5 \times 10^5$ | 0.1177 | 0.1141 | -3.1% |
| $Re_c = 1.5 \times 10^6$ | 0.1185 | 0.1158 | -2.3% |
| $Re_c = 2.5 \times 10^6$ | 0.1205 | 0.1183 | -1.8% |

## 3 Results and discussion

### 3.1 Validation

Initial results can be validated against existing experimental results. To the best of the authors' knowledge, the only experimental dataset available concerning the aerodynamic performance of the FFA-W3-211 airfoil is Bertagnolio et al. (2001). In his work, the FFA-W3-211 airfoil was tested for a chord-based Reynolds number of $Re_c = 1.8 \times 10^6$ in the low-speed wind tunnel L2000 at KTH. The experimental facility has a 2 by 2 metres section, with a turbulence intensity of $0.15\%$. The static polars were acquired for $-5.43° < \alpha < 31°$. Finally, results from the compressible panel method from RFOIL simulations are included (Van Rooij, 1996). Results shown in figure 7 show great agreement in both lift and drag coefficients and allow the wind tunnel model and methodology to be validated.

### 3.2 Static polars

The newly acquired airfoil polars are the starting point for this analysis. Static results are presented in Figure 8. The lift coefficient linear region slope slightly increases with the Reynolds number. Here, the highest aerodynamic efficiency shows significant Reynolds number effects. Firstly, the $(C_l/C_d)_{max}$ increases in magnitude with increasing Reynolds numbers. Secondly, the value of $\alpha$ at which the highest aerodynamic efficiency is found decreases with increasing Reynolds numbers. The aerodynamic efficiency is found to be heavily dependent on the Reynolds number, as explained in Ceyhan et al. (2017).

A regime was identified after $Re = 2 \times 10^6$, after which the maximum lift coefficient value increases. The same regime was identified in Brunner et al. (2021), highlighting a shift from trailing-edge stall to leading-edge stall for higher Reynolds numbers. The angle of attack at which the maximum lift coefficient is found follows a similar trend. At low Reynolds numbers, stall occurs at higher angles of attack, which decrease as the Reynolds number increases due to a combination of different boundary layer thicknesses and separation points. It can be concluded that a different regime is identified, which is solely due to the effects of the Reynolds number.

The effects of a laminar separation bubble (LSB) can be observed in the pressure distribution for the low Reynolds cases. This is particularly strong for $Re_c = 5 \times 10^5$, where the bubble can be observed on both the suction and pressure sides in




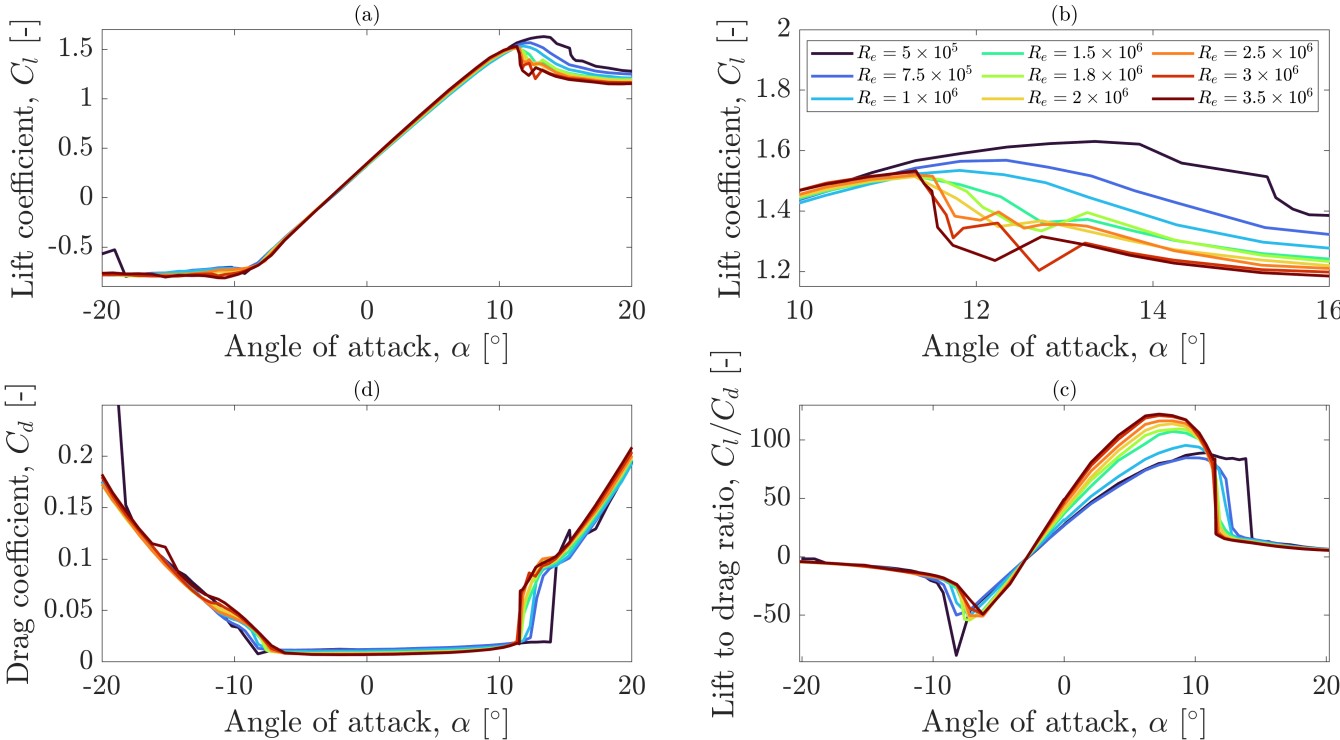

**Figure 8.** Clean static polars for the FFA-W3-211 for $5 \times 10^5 \le Re_c \le 3.5 \times 10^6$.

Figure 9 a. As the Reynolds number increases, the effects of the LSB decrease. The LSB fixes the flow transition to around $x/c = 30\%$, decreasing in magnitude as the flow inertia becomes greater.

The position and magnitude of the suction peak change with the Reynolds number. A higher suction peak is first identified for low Reynolds numbers. For the same angle of attack, the $x/c$ position at the highest magnitude slightly shifts towards the trailing edge for increasing Reynolds numbers. The trend is shown in figure 10, where static stall angle is also presented. Here, the magnitude of the suction peak shows Reynolds number effects.

The different stall behaviour also highlights the effects of the Reynolds number. The stall behaviour is gradual for the regimes in which an LSB is found in the pressure distribution. The stall becomes more abrupt for the highest Reynolds number, in agreement with high-Reynolds literature. The stall can be classified as a trailing-edge stall for all the tested Reynolds numbers and angles of attack. As the Reynolds number increases, the separation point on the suction side moves towards the leading edge. Figure 9 c shows that for $\alpha = 13.3°$, the flow is only attached for for the $Re_c = 5 \times 10^5$ case. Figure 9 d shows the effects of the Reynolds number on the location of the separation point, which moves upstream with increasing Reynolds numbers. This is associated with a decreasing suction peak magnitude, $C_{p_{min}}$, as shown in figure 10 b. Figure 10 a shows the Reynolds effects on the maximum lift coefficient. As shown in Brunner et al. (2021) and Kiefer et al. (2022) for a NACA0021,





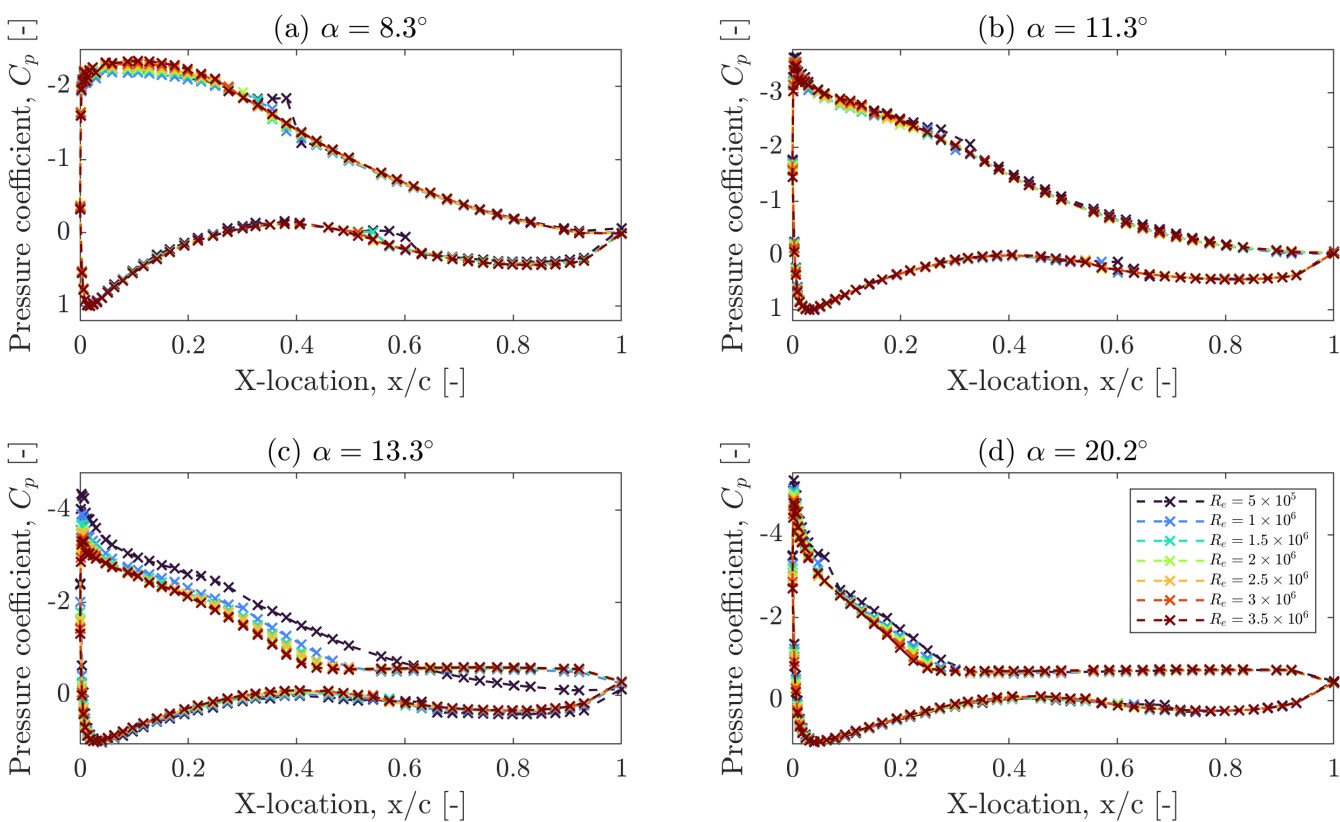

**Figure 9.** Pressure contours for relevant angles of attack.

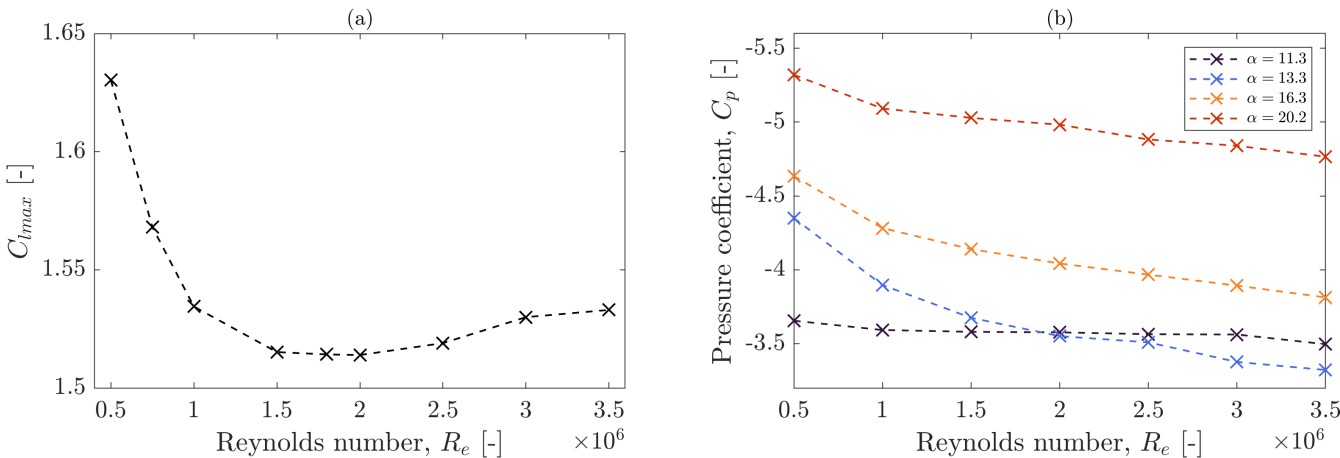

**Figure 10.** Maximum lift coefficient (a) and suction peak (b) as a function of Reynolds number and angle of attack for the clean airfoil.



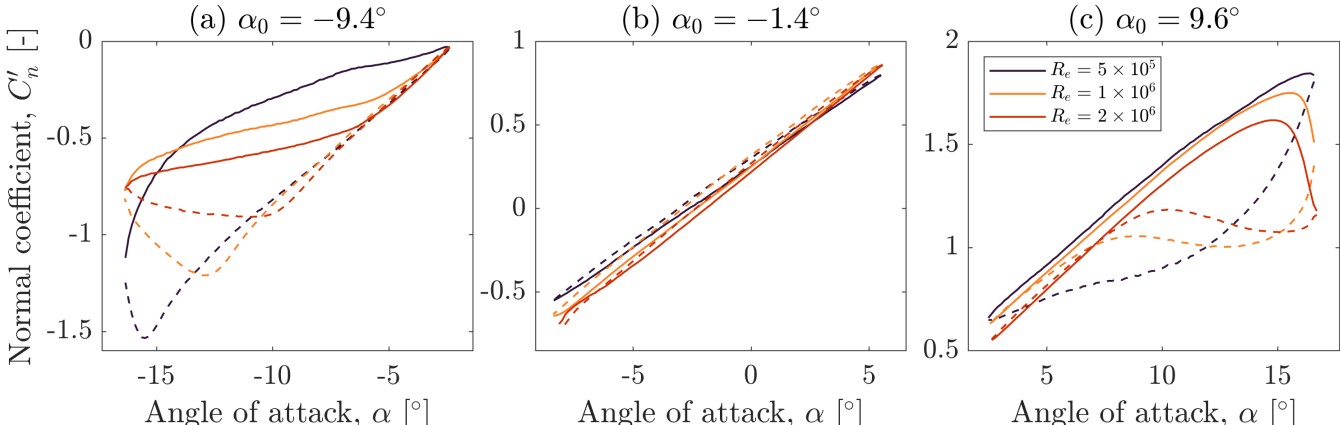

**Figure 11.** Reynolds number sweep for $f = 1.2Hz, A = 7°$. The reduced frequency range is $k = 0.182, 0.091, 0.046$. The full line represents the upstroke wing movement, while the dotted line represents the downstroke wing movement. Data averaged over 50 dynamic stall cycles.

the maximum lift coefficient value is found to increase past the $Re_c \approx 2 \times 10^6$ regime.

## 3.3 Dynamic polars

280 As previously mentioned, capturing the large wake shed by the airfoil with a rake for the dynamic cases was impossible. Therefore, the normal coefficient results are presented and averaged over 50 loops of dynamic stall for all cases. The results are divided into Reynolds number, reduced frequency, oscillating frequency and amplitude sweeps for easy comparison. The normal coefficient results are presented with an accuracy of $\pm 1°$ due to some minor play in the linear actuator motion, particularly found for the highest inertia cases.

### 285 3.3.1 Reynolds number effects

Results in Figure 11 show the Reynolds number effects on the dynamic normal coefficient. Three representative mean angles are chosen as the centre of the sinusoidal oscillation in the mean, positive and negative regions of the polars. The combined effects of the Reynolds number and reduced frequency can be isolated, starting with the positive regime. A delay is observed in the angle of attack at which stall occurs, which is inversely proportional to the Reynolds number. This is due to the higher

290 inertia introduced in the system, at which the reduced frequency decreases and with decreasing free stream velocity. The normal coefficient magnitude decreases with increasing Reynolds numbers and reduced frequencies to the point where, for highly unsteady reduced frequencies, separation does not take place for angles of attack as high as $\alpha = 16.6°$
The linear region of the polars shows minor differences in magnitude. A difference in slope is found as both parameters are varied, in agreement with previous literature (De Tavernier et al., 2021; Kiefer et al., 2022). The linear region slope is found to





**Table 3.** Reduced frequencies range matrix for the dynamic oscillations.

| / | $Re = 5 \times 10^5$ | $Re = 10^6$ | $Re = 2 \times 10^6$ |
|---|---|---|---|
| $f = 0.6Hz$ | 0.091 | 0.046 | 0.023 |
| $f = 1.2Hz$ | 0.182 | 0.091 | 0.046 |
| $f = 1.8Hz$ | 0.273 | 0.137 | 0.068 |
| $f = 2.4Hz$ | 0.365 | 0.182 | 0.091 |

increase with decreasing reduced frequencies and increasing Reynolds numbers.

Lastly, the negative region of the polars showcases two different reattachment regimes. Here, similarly to the positive region, the highest reduced frequencies and Reynolds numbers keep the boundary layer attached at lower angles of attack. At the end of the oscillation, dynamic stall occurs abruptly, leading to a drastic increase in normal coefficient and a rapid reattachment mechanism, significantly faster than the equivalent at lower reduced frequencies.

### 3.3.2 Reduced frequency

Together with the Reynolds number, the reduced frequency directly affects the airfoil performance. Given the low Mach number, it is possible to isolate the effects of $k$ by varying the oscillating frequency, described in Eq. (4) and Reynolds number. It was only possible to test one reduced frequency due to setup limitations, corresponding to a value of $k = 0.091$ as seen in Table 3, which falls in the unsteady regime according to Corke and Thomas (2015).

The results are shown in figure 12. Except for a decreasing force magnitude in upstroke results for the linear and positive regions, these two seem relatively independent of the Reynolds number. However, the two regions show great differences for the same reduced frequency. Firstly, the nose-down reattachment in figure 12 C exhibits different magnitudes and behaviours. For the lowest Reynolds number and frequency, the reattachment is quite linear. As both parameters double, the initial reattachment is faster, with a strong and sudden decrease in loads. The last case, for $Re_c = 2 \times 10^6$ and $f = 2.4Hz$, bridges between the two previous cases. Secondly, the negative region of the polars shows clear Reynolds numbers effects in the negative stall region. These are visible in the lower minimum normal coefficient attained at higher Reynolds numbers. All cases show a smooth negative stall, with no sudden decrease in loads as observed in the positive static region due to the airfoil's camber. Finally, the nose-up motion in the negative region shows a consistent reattachment trend, which can be considered independent of the Reynolds number and oscillating frequency. The mechanisms that describe the boundary layer attachment are a function of the variation in leading-edge suction peak and trailing-edge stall evolution.

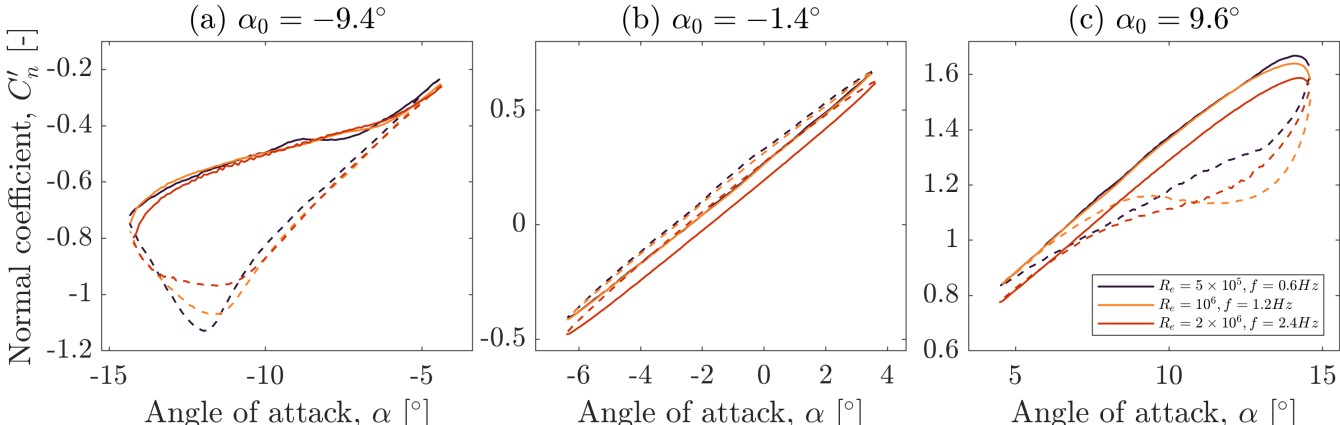

**Figure 12.** Reduced frequency trend for a constant value of $k = 0.091$, covering different Reynolds numbers and oscillating frequencies. The full line represents the upstroke wing movement, while the dotted line represents the downstroke wing movement. Data averaged over 50 dynamic stall cycles.

### 3.3.3 Oscillating frequency effects

The effects of oscillating frequency are shown in figure 13, divided by mean angle and Reynolds number. The lowest frequency displays a significantly faster reattachment behaviour for all the tested cases. For the lowest reduced frequency of $k = 0.23$, the flow can be considered quasi-steady Corke and Thomas (2015), implying that the flow remains attached for slightly more than

its static equivalent case. The peak normal coefficient for the positive mean angles is delayed by a few degrees and displays a higher magnitude than the static equivalent, with the value increasing as the frequency increases for all Reynolds numbers.

Finally, the oscillating frequency is found to influence the linear region slope. The force slope decreases with increasing instability. Here, a Reynolds number dependency is also observed, for which the upstroke and downstroke difference in time-averaged normal coefficient exhibits fewer differences as the Reynolds number increases. Following the conclusions of some

high-Reynolds numbers literature Kiefer et al. (2022), it is expected that the thinner boundary layer will progressively reduce the upstroke and downstroke difference in pressure distribution. Stall is classified as trailing edge, with the separation point fastly moving upstream of the airfoil until the flow is fully detached from the airfoil, depending on the oscillating amplitude. One interesting effect is noted here. Except for one, all the presented linear region cases exhibit a stronger normal coefficient magnitude during the downstroke phase. The only case this does not occur is for the highest tested reduced frequency of

$k = 0.364$. Here, due to the higher flow instability, the airfoil's upstroke motion shows stronger loads.

### 3.3.4 Amplitude effects

Results in figure 14 display the effects of the oscillating amplitude for a Reynolds number of $Re_c = 10^6$. While pitching the airfoil around the same mean angle, it is immediately visible how, for the positive and negative regions of the polars, the

**Figure 13.** Oscillating frequency sweep for $Re_c = 2 \times 10^6$, $A = 7°$, covering the quasi-steady, unsteady and highly unsteady regimes. The full line represents the upstroke wing movement, while the dotted line represents the downstroke wing movement. Data averaged over 50 dynamic stall cycles.





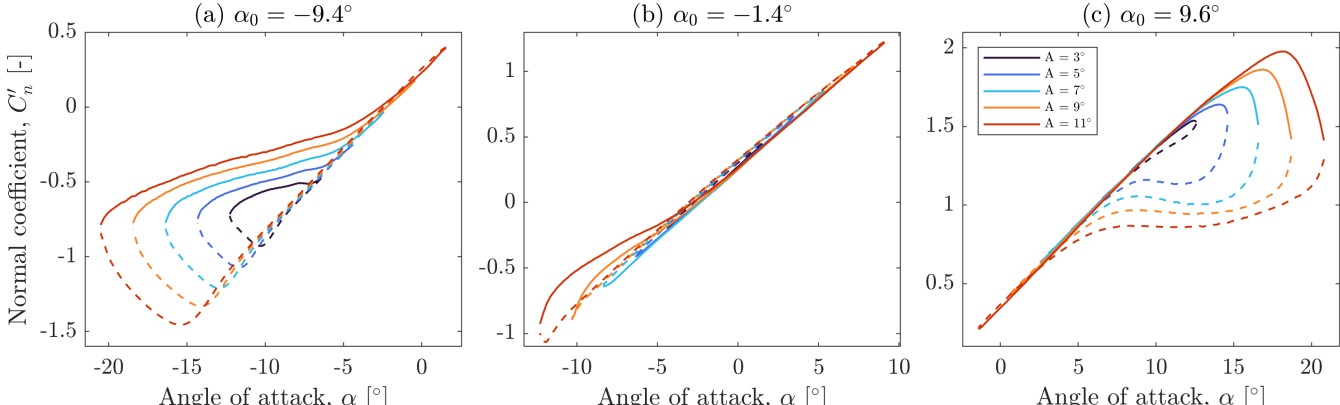

**Figure 14.** Oscillating amplitude sweep for $Re_c = 1 \times 10^6$, $f = 1.2Hz$, covering the quasi-steady, unsteady and highly unsteady regimes. The full line represents the upstroke wing movement, while the dotted line represents the downstroke wing movement. Data averaged over 50 dynamic stall cycles.

dynamic loop shape is similar between all cases and changes linearly with the angle of attack. For the linear region case,

the highest oscillations can already capture the effects of dynamic stall in the lower regime, showing a difference in normal coefficient for the same angle of attack. This is particularly visible for the $A = 11°$ case, where the oscillation is large enough to capture the dynamic stall loop. As expected, there are differences between the positive and negative regions of the polars due to the airfoil's camber. Due to the cambered nature of the FFA-W3-211 airfoil, a larger dynamic stall loop is associated with the negative region of the polars where the convergence to the linear region is slow and displays greater discrepancies in

the force magnitude. One example is the flow reattachment behaviour, which is observed to be linear in the negative region, while this exhibits a force increase in the positive region.

### 3.4 Dynamic pressure coefficient

Similar to the static counterpart, the dynamic pressure coefficient was retrieved through integration. Given the airfoil oscillation, these were averaged for an instantaneous angle of attack with a tolerance of $\pm 0.1°$. A number of taps were excluded from the

analysis as these were found to be partially blocked, slow or unplugged. As a result, the following plots are obtained from a pressure distribution of 83 pressure taps.

Results are presented for angle of attack of $\alpha = -13°, 14°$. Due to the pressure scanner range, the average pressure distribution exhibits a coarser instantaneous behaviour at low Reynolds numbers.

As this increases, the data appears smoother also thanks to the lower reduced frequency. The LSB observed in the static

results is present for $Re = 5 \times 10^5, 10^6$ cases, and its magnitude decreases as the Reynolds number increases. It can also be noted that the pressure distribution values are not identical and significantly vary for the upstroke and downstroke motions. The negative region of the polars showcases a clear suction peak in the downstroke motion, which collapses as the airfoil is pitched





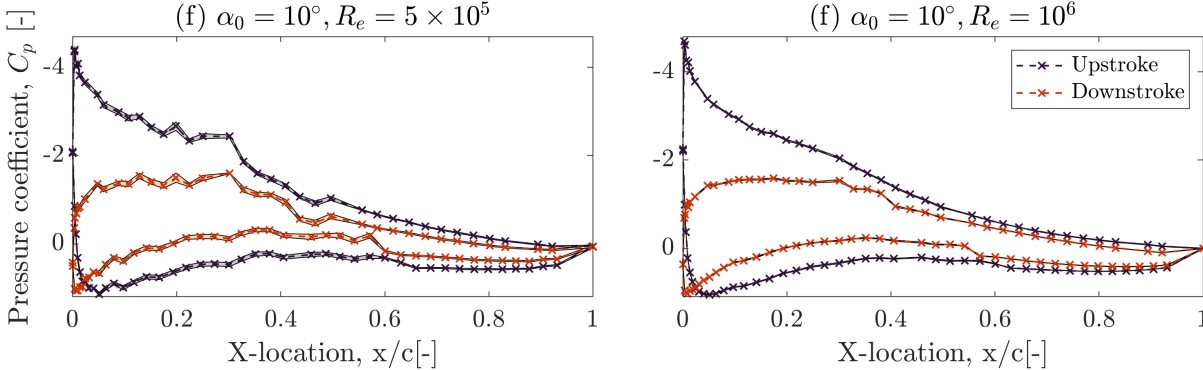

**Figure 15.** Pressure coefficient standard deviation computed for $A = 7°$, f $= 1.2 Hz$

up. Similar to the static case, this depends on the Reynolds number and reduced frequency combination, and it collapses as the Reynolds number increases. This is partially mitigated by the reduced frequency term. As the flow reattaches, the trailing edge
provides the lift, which presents the highest pressure difference. The hysteresis loop is caused by the unsteady flow effects. In the deep dynamic stall conditions, unsteady effects are predominant, with higher force values obtained at higher angles of attack. As highlighted in Kiefer et al. (2022), two mechanisms are responsible for the load development. This is driven by the increase in suction peak magnitude near the leading edge and the boundary layer separation mechanisms originating at the trailing edge. The magnitude variation of the suction peak is responsible for the lift increase.

## 4 Conclusions

This work describes the experimental aerodynamic results of the FFA-W3-211 airfoil. The wind energy community widely uses the airfoil as a popular reference. An experiment was performed in the low-speed, low-turbulence tunnel at the Delft University of Technology, for which a new wing model with a chord length of $c = 600m$ and height of $h = 1246mm$ has been manufactured. Previously, several wing geometries have been tested in this facility. The airfoil has been pitched around its
quarter chord point sinusoidally to acquire the first dynamic data for the airfoil geometry for chord-based Reynolds numbers of up to $Re_c = 2 \times 10^6$.

The static data highlights the presence of a laminar separation bubble that disappears with increasing Reynolds numbers. This flow feature is responsible for the shift in static stall behaviour. Similarly to the collapse of the LSB, the suction peak magnitude and separation point exhibit strong dependency on the Reynolds number.
The dynamic results display the effects of the individual variation of inflow and oscillating parameters. The reduced frequency and Reynolds number effects have been described, highlighting the influence on dynamic stall, reattachment and hysteresis loops. No particular fundamental behaviour is observed for increasing oscillating amplitudes. The upstroke and

**Figure 16.** Dynamic pressure coefficient for $A = 7°$, f $= 1.2Hz$



downstroke pressure distributions showcase fundamental differences in flow structure development, ultimately affecting the overall forces balance.

*Data availability.* Data can be downloaded under CC BY 4.0 license at 10.4121/92716ecf-b075-41a1-ab93-8e2af785a404

**Appendix A**

An initial shift of $\alpha = 1.4°$ was observed in the dynamic data. As a result, a calibration was carried out against both static and dynamic ultra-low frequency data. This has allowed to quantify the shift, which is consistent for most cases for $Re = 5 \times 10^5, 10^6$ and the negative mean angles for $Re = 2 \times 10^6$. However, in some occasions, the data shift observed in the positive

regime for the highest Reynolds number shows some irregularities which fall in the $\pm 1°$ regime. This has to be assumed as the reported experimental error for the pitching airfoil cases.

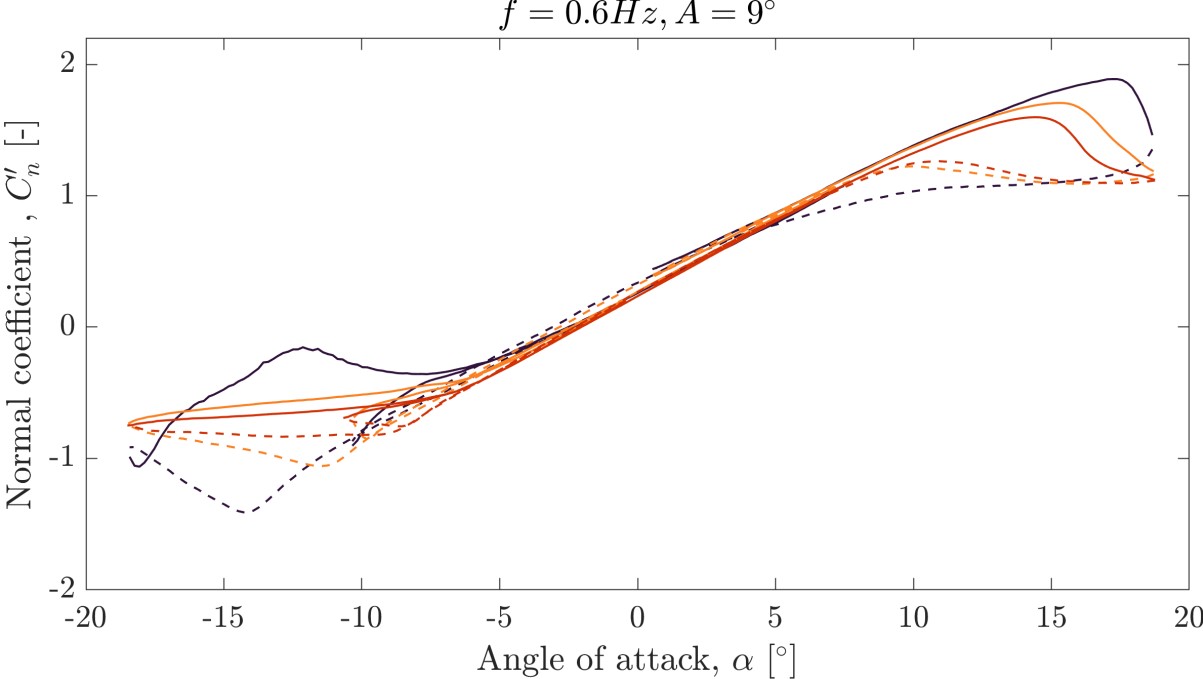

**Figure A1.** Data overlap between the negative, mean and positive regimes for $A = 9°$, f = $0.6Hz$.





*Author contributions.* SC was responsible for the overall research, under the supervision of DDT and DvT. SC carried out the experimental measurements as well as the post-processing, visualization and analysis of the measurements.

*Competing interests.* The authors declare that they have no conflict of interest.



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
