# Peer review of "The experimental characterisation of dynamic stall of the FFA-W3-211 wind turbine airfoil"

_Wind Energy Science, 2025_

## Referee Comment (RC1)

Review of https://doi.org/10.5194/wes-2025-121
Date: 08/08/2025

*Note: The authors are encouraged to have a critical look at the inconsistencies of the whole paper and not only improve the specific points mentioned here.*

Line 75 Reference of Gardner et all (2023) is mentioned without giving context. Please summarize its findings

Around Line 80: Mainly focused on the experimental data and its shortcomings for Beddoes-Leishman model. However, other models such as Onera, Snel (and improved version from Adema et all (2020)), Oye are also still used during wind turbine designs. How these experimental results could support these models should be included.

Although the reason becomes clear when reading the rest of the paper, please clearly motivate the inclusion of the literature about the Re number effects investigation. Also there are inconsistencies such as in Line 51 mentioned 2 airfoils tested, and then listed three airfoils...

A general question about the model checks: What is the impact of the model surface deformation due to uneven distribution of the loads? In other words, please comment if and how the model flexibility effects the results.

[Figure]

 How do these red circled bumps or gaps impact the (drag) measurement?

Line 137 why only -20 and 20 and not much larger angles? I would expect at least between -25 and 25 but considering previous experiments also from -60 and 60.

The actuator calibration part is not very clear:
Line 166 The increased aerodynamic loads for the clean airfoil configuration shifted to the Zero angle of attack. Do you mean zero lift angle of attack?

Not sure where 0.5mm difference in initial arm position between transitioned Re=2x106 and clean Re=1x106 come from? A diagram of the system showing this angle might help here. The last point is whether this calibration was finally implemented and how?

Not sure if Figure 4 contributes to anything.

Line 178, adding k here helps the reader ... reduced frequency, k, which is defined...

Line 220 what traditional voices? Which last paragraph? This part is not clear

Line 234 says Table 2 shows the slopes, but its caption says calibration. Was this difference introduced as calibration?

Line 246 what is a compressible panel method? Or do you mean compressibility corrected panel method

Line 247 I am not sure if I would call this great agreement by looking at these two graphs. For the purpose of this paper, it would be better to show Cl alpha graph to visualize the actual differences in relation with angle of attack instead of cl/cd which is more relevant as a design parameter. If there is pressure distribution comparison or Cm, these would also be added value.

Line 255 here you are stating that after re=2x106, clmax start increasing and it is a different regime. Using the Reference, you also imply that this difference is the shift from trailing edge stall to leading edge stall. However, later Line 271, all stall events are stated as trailing edge stall for all tested Re numbers. This is not only contradictory to initial observation but also lacks supporting arguments. And it is likely not correct for the static stall cases. Please support these arguments using pressure distributions.
.
Section 3.3.1 needs to be reconsidered first and rewritten. Figure 11 is hard to follow since it doesn't show reduced frequencies. According to the results, Re effects and reduced frequency effects are not isolated; they are combined and the author should be careful when mentioning one without the other.
Another observation is the significant Re number impact and its difference between the upstroke and downstroke movements which is remarkable however not covered.
Line 296 when the negative region of the polars mentioned as last but the figure is included in the beginning of Figure 1, it is very confusing to follow this. Please rearrange either the figures or the interpretation part.
Figure 11: Please include the reduced freq values directly here, next to the Re numbers.

Line 305 the claim that linear and positive regions being independent from Re effects is only true for the upstroke movement and not for the downstroke.

Line 307 Nose down reattachment - does it mean reattachment in the downstroke movement?

Figure 12C: here it seems like Re=1x106 in the downstroke movement is distinctly different from 0.5 and 2x106 cases. In all other cases in the positive region, there is no such distinct behavior. What can be the explanation of this?
Figure 12: please add amplitude value

Line 323 which instability?

Line 326 how could the separation point's fast movement and its relation with trailing edge stall be concluded from these graphs? Also see the previous comment on the static stall type which is conflicting. It could be trailing edge stall, but the reference you refer to, and a few of the pressure distribution comparisons do not fully support this.

Line 328 if the linear case is the middle one, there is hardly any difference.

Line 330 Please either include this reduced freq value of 0.364 to the test results and the test matrix or delete this sentence.

Section 3.3.4 this section should also be improved considering the comments on the previous sections (linear region or positive region?, flow reattachment in downstroke or upstroke, etc...)

Section 3.4 Dynamic pressure distribution is a more appropriate title for this one.

There is no reference to Figure 15 or 16 making readingand following arguments very difficult.

Line 349 presence of LSB is mostly speculative if only the Figure 15 is concerned.

Figure 16: it is quite difficult and even speculative to claim the presence of the LSB on the lower Re numbers by looking at these figures due to these discrepancies in the pressure readings. It could help if you add static polars for the same cases.

Line 349 "As this increases"... - what is referred here?

Line 352 how do the polars show clear suction peak? Polars only show loads.

Line 355 where is the detachment and reattachment?

Line 359 if the suction peak magnitude is responsible for the lift increase which is correct, which case is referred to in line 355 where trailing edge provides the lift.

The drag and moment data and results from zigzag tape are missing.

---

## Author Comment (AC2)

**The experimental characterisation of dynamic stall of the FFA-W3-211 wind turbine airfoil Response to Referee report 2**

**Simone Chellini**

**October 2025**

The authors appreciate the kind words of the reviewer and share the view that unsteady aerodynamics is important for wind turbine performance and root-cause analyses, in particular considering thicker airfoils and larger angles of attack. The reviewer provided some suggestions for textual improvements. The authors have implemented all. These are listed here again for completeness:

- Comment 1: "... is only attached for for the  $Rec=510^5$  case ...". Duplicate "for for" should be corrected as Single "for"
- Comment 2 "... the stall is classified as trailing edge, with the separation point fastly moving upstream ...". "fastly" is nonstandard or archaic. Probably "quickly" or "rapidly" may be better technically.
- Comment 3 "... with a chord length of c=600m and height of h=1246mm...". Missing space or unit identifier: "600 m" suggests 600 meters which is obviously wrong; they mean 600 mm. Probably a typo: "600 mm".

---

## Author Comment (AC3)

**The experimental characterisation of dynamic stall of the FFA-W3-211 wind turbine airfoil Response to Referee report 1**

**Simone Chellini**

**October 2025**

The authors thank the reviewer for the valuable feedback. We appreciate the recognition of our work for the importance and quality of the data set and the opportunity to improve our manuscript. The main concerns of the reviewer and our response are summarized below.

- The results section needs general improvements, and some specific points in the data analysis and interpretation require attention. We agree and will revise accordingly. Specifically mentioned points are addressed in more detail further below.
- Some data were introduced but not reported in the results section, for example, tripped cases and other measured quantities. Indeed, this was the case; therefore, we removed the reference to these data that are not ready and not necessary to be included here.
- The discussion of the literature in the introduction was found to be not always clear and lacked completeness in the eye of the reviewer. Regarding the clarity issues raised, changes were made accordingly (see below) and some additional references were added. However, it is important to understand that our literature discussion was designed to highlight the importance of the FFA-W3-211 airfoil for reference wind turbines and to focus on the lack of experimental data. Some flow physics, like dynamic stall and related Reynolds number effects, were introduced to provide context why certain quantities were measured and why certain parameters were varied. For the same reason, one particular dynamic stall model, the Beddoes-Leishman model, was referenced to provide a prominent example. The literature discussion was, by no means, meant to be comprehensive regarding other airfoils, general flow physics or the range of available dynamic stall models. In the final manuscript, we will streamline the literature discussion to make these points more clearly.

Comments addressing specific points and our replies can be found in the following.

• Comment 1: Line 75 Reference of Gardner et al. (2023) is mentioned without giving context. Please summarise its findings.

**Response:** Additional explanation was included in the final manuscript summarising the findings of Gardner et al.

• Comment 2: Around Line 80: Mainly focused on the experimental data and its shortcomings for Beddoes-Leishman model. However, other models such as Onera, Snel (and improved version from Adema et al. (2020)), Oye are also still used during wind turbine designs. How these experimental results could support these models should be included.

Response: We agree that other dynamic stall models are in use for wind turbine applications and could benefit from calibration with the reported data. We have focused the literature on the baseline (Beddoes-Leishman) model as an illustration. However, we have expanded the paragraph to highlight the importance of other dynamic stall models, making the story more complete.

• Comment 3: Although the reason becomes clear when reading the rest of the paper, please clearly motivate the inclusion of the literature about the Re number effects investigation. Also there are inconsistencies such as in Line 51 mentioned 2 airfoils tested, and then listed three airfoils...

**Response:** We agree that the literature provided about the Reynolds number effects is not necessarily needed here, particularly the work focusing on static measurements only. Therefore, we have decided to follow the reviewer's suggestion and remove the paragraph from line 51 to line 62.

• Comment 4: A general question about the model checks: What is the impact of the model surface deformation due to uneven distribution of the loads? In other words, please comment if and how the model flexibility effects the results.

Response: During the test campaign, no surface deformations were detected. While this was only visually inspected, with the weakest point being the trailing edge, we believe that the model can be assumed to be rigid due to the relatively high carbon fibre thickness and local refinements. Small aeroelastic deformations of the setup were identified but handled with care in the control calibration of the set-up and in the post-processing where needed (see comment 7). This will be more clearly highlighted in the revised manuscript.

• Comment 5: How do these red circled bumps or gaps impact the (drag) measurement?

Response: The light colored areas, shown at the top of Fig. 3a and b, are due to light reflections on the wind tunnel windows. The dark colored bump on Fig. 3b, only present for the higher Reynolds number, shows a small imperfection on the airfoil surface. This will indeed affect the loads (particularly drag) at the specifically shown angle of attack and Reynolds number, but this is fairly small. The pressure tabs are installed far away from this imperfection, and drag measurements are averaged over a span range of 20 cm. Therefore, we believe that this effect is only marginal. At higher angles of attack, the boundary layer thickens, and the flow's sensitivity to this small imperfection is no longer visible. To illustrate the variation of the drag along the airfoil span, Figs. 1 and 2 are provided. Here, it is clear that although the drag varies along the span, the variations are considered small and within an acceptable range, allowing us to assume that the model is (more or less) 2D. This will be clarified in the revised manuscript.

Figure 1: Wake rake-acquired drag coefficient over height span for  $\alpha = 0, 6, 9, 12$  deg, Re =  $10^6$ .

Figure 2: Pressure deficit in the wake as obtained by the wake for  $\alpha = 0$ , Re =  $10^6$ .

• Comment 6: Line 137 why only -20 and 20 and not much larger angles? I would expect at least between -25 and 25 but considering previous experiments also from -60 and 60.

Response: While practically the set-up would allow for measuring a larger angle of attack range, changes would need to be made manually to the set-up for larger angle of attack deflections. The time available for this test campaign in the wind tunnel was limited, and therefore it was decided to limit the angle of attack variation from -20 to 20deg. Furthermore, blockage effects start to become significant for this test setup at angles of attack higher than 20deg. While for the static polars, blockage corrections can be implemented, for the dynamic measurements these corrections are not proven nor implemented in this work.

• Comment 7: The actuator calibration part is not very clear: Line 166 The increased aerodynamic loads for the clean airfoil configuration shifted to the Zero angle of attack. Do you mean zero lift angle of attack?

**Response:** Due to the aerodynamic loads acting on the airfoil model and pitching system, the set-up was experiencing some aeroelastic response. To cope with this, the actuator calibration (linking the actuator motion to the airfoil angle of attack) was performed separately for different Reynolds numbers. We agree that this is not clearly described in the manuscript. For clarity, we propose to simplify the explanation in the revised manuscript.

• Comment 8: Not sure where 0.5mm difference in initial arm position between transitioned Re=2x106 and clean Re=1x106 come from? A diagram of the system showing this angle might help here. The last point is whether this calibration was finally implemented and how?

**Response:** This comment is closely linked with comment 7. Similarly, we propose to simplify the explanation of the actuator calibration and clarify how this is implemented.

• Comment 9: Not sure if Figure 4 contributes to anything.

**Response:** We agree with the reviewer and propose to remove Fig. 4 from the final manuscript.

• Comment 10: Line 178, adding k here helps the reader ... reduced frequency, k, which is defined...

**Response:** Agreed. The suggestion was implemented in the revised manuscript.

• Comment 11: Line 220 what traditional voices? Which last paragraph? This part is not clear

**Response:** Here we meant the Allen and Vincenti corrections adopted in static testing. As this wording was unclear and the content not crucial to the paragraph, we removed this part from the text.

• Comment 12: Line 234 says Table 2 shows the slopes, but its caption says calibration. Was this difference introduced as calibration?

Response: Due to set-up limitations for the dynamic testing, the model cannot span exactly from the bottom wall up to the upper wall. A small gap is present between the end of the model and the walls, allowing the model to rotate freely. This small gap introduces extra 3D effects to the model set-up. Additionally, for dynamic testing, a slot has been incorporated into the wind tunnel floor to accommodate the movement of pressure taps as the model rotates. This gap causes some mass outflow. To quantify these effects on the lift measurements, a comparison is made between the static lift polar with and without taping these gaps. This comparison is shown in Figure 6b and quantified in Table 2. The difference is communicated in the manuscript but it is not introduced in any calibration or correction procedure. We have emphasised this further in the manuscript, amended the table's caption to avoid confusion and improved the table's clarity in the final manuscript.

 Comment 13: Line 246 what is a compressible panel method? Or do you mean compressibility corrected panel method

**Response:** We indeed mean a panel method with compressibility corrections. We adjusted this sentence for clarity.

• Comment 14: Line 247 I am not sure if I would call this great agreement by looking at these two graphs. For the purpose of this paper, it would be better to show Cl alpha graph to visualize the actual differences in relation with angle of attack instead of cl/cd which is more relevant as a design parameter. If there is pressure distribution comparison or Cm, these would also be added value.

Response: The reviewer is right to say that a "great" agreement is unnecessarily exaggerated. We corrected this in the final paper. Also, we agree that a Cl-alpha curve would provide interesting insights, more than the Cl/Cd-alpha plot shown right now. Therefore, we propose to replace Figure 7b. Unfortunately, no pressure distribution data is available from Bertagnolio et al. (2001) for comparison.

• Comment 15: Line 255 here you are stating that after re=2x106, clmax start increasing and it is a different regime. Using the Reference, you also imply that this difference is the shift from trailing edge stall to leading edge stall. However, later Line 271, all stall events are stated as trailing edge stall for all tested Re numbers. This is not only contradictory to initial observation but also lacks supporting arguments. And it is likely not correct for the static stall cases. Please support these arguments using pressure distributions.

**Response:** As the Reynolds number increases, the separation point moves upstream as seen in Figure 9c. By looking at the pressure distribution and stall behaviour from the Cl vs AoA plot, we can assume the transition shifts from trailing-edge stall to mixed stall. After additional analysis, we have amended the final manuscript with this conclusion, including the transient pressure distribution behaviour. We have also amended line 271, removing the reference to all-trailing-edge stall.

• Comment 16: Section 3.3.1 needs to be reconsidered first and rewritten. Figure 11 is hard to follow since it doesn't show reduced frequencies. According to the results, Re effects and reduced frequency effects are not isolated; they are combined and the author should be careful when mentioning one without the other. Another observation is the significant Re number impact and its difference between the upstroke and downstroke movements which is remarkable however not covered. Line 296 when the negative region of the polars mentioned as last but the figure is included in the beginning of Figure 1, it is very confusing to follow this. Please rearrange either the figures or the interpretation part. Figure 11: Please include the reduced freq values directly here, next to the Re numbers.

Response: Thank you for your comment. After carefully reviewing the manuscript, we identified that the reduced frequency and Reynolds numbers were not consistently mentioned throughout the paper. We have corrected this at all relevant locations. We believe it is insightful to keep the sections on reduced frequency and Reynolds numbers separated (section 3.3.1 and 3.3.2), although the effects are not entirely isolated. We rewrote section 3.3.1 with the purpose of improving readability and consistency. As an example, the reviewer requested that the interpretation in the final manuscript be rearranged to follow the figure. We also added a paragraph discussing the significant impact of the Reynolds number in the upstroke movement.

• Comment 17: Line 305 the claim that linear and positive regions being independent from Re effects is only true for the upstroke movement and not for the downstroke.

Response: Thank you for your comment, we amended the paragraph.

• Comment 18: Line 307 Nose down reattachment - does it mean reattachment in the downstroke movement?

**Response:** We indeed referred to the reattachment in the downstroke movement. We have amended the final manuscript.

• Comment 19: Figure 12C: here it seems like Re=1x106 in the downstroke movement is distinctly different from 0.5 and 2x106 cases. In all other cases in the positive region, there is no such distinct behaviour. What can be the explanation of this? Figure 12: please add amplitude value

**Response:** It is rightly pointed out by the reviewer that from Fig. 12c, it looks like the downstroke movement at  $Re = 1x10^6$  is distinctly different from Re = 0.5 and  $2x10^6$ . This is an interesting observation. When considering Figure 13, it is clear that the combination of oscillating frequency and Reynolds number determines the sudden reattachment of the boundary layer. Figure 13 shows that all cases are consistent with each other for frequency ranges, with clear trends in downstroke reattachment identifiable for all the positive mean angles. The discrepancy in figure 12C may possibly be explained as the result of Reynolds number effects.

• Comment 20: Line 323 which instability?

**Response:** "Instability" was not the appropriate word choice in this sentence, it was meant to be "reduced frequency." The clause was dropped as it was not needed.

• Comment 21: Line 326 how could the separation point's fast movement and its relation with trailing edge stall be concluded from these graphs? Also see the previous comment on the static stall type which is conflicting. It could be trailing edge stall, but the reference you refer to, and a few of the pressure distribution comparisons do not fully support this.

**Response:** The reviewer has a good point here. We will adjust the statements for consistency and point out that the explanation is not conclusive in the final manuscript.

• Comment 22: line 328 if the linear case is the middle one, there is hardly any difference.

**Response:** The linear case is shown in Fig. 13B, where the highest frequency of 2.4 Hz combined with the lowest free stream, returns the highest reduced frequency. It can be observed from the normal coefficient plot that the force magnitude trend is inverted with respect to all the other cases in Fig. 13E and 13H.

• Comment 23: Line 330 Please either include this reduced freq value of 0.364 to the test results and the test matrix or delete this sentence.

Response: We removed the sentence starting at line 328 in the revised manuscript.

• Comment 24: Section 3.3.4 this section should also be improved considering the comments on the previous sections (linear region or positive region?, flow reattachment in downstroke or upstroke, etc...)

**Response:** We acknowledge this comment and will make sure that the final manuscript is more consistent. We will rewrite the section to improve readability, remove inconsistencies and strengthen our arguments when hypothesising flow behaviour.

• Comment 25: Section 3.4 Dynamic pressure distribution is a more appropriate title for this one.

**Response:** Agree. We amended the final manuscript accordingly.

• Comment 26: There is no reference to Figure 15 or 16 making reading and following arguments very difficult.

Response: We amended the final manuscript and explicitly integrated Figures 15 and 16 in the discussion.

• Comment 27: Line 349 presence of LSB is mostly speculative if only the Figure 15 is concerned.

Response: We agree with the reviewer that it is hard to identify the presence of laminar separation bubbles with certainty from the pressure distributions in Fig. 15 (and 16). Particularly at the lower Reynolds numbers, the discrepancies in the pressure are significant, making it hard to distinguish a LSB. However, at the higher Reynolds numbers, the pressure distributions are smoother. Here, the pressure distributions hint towards the presence of an LSB, particularly supported by the fact that the x/c position is consistent with the static (simulated) data for a Reynolds number of 106. We must clearly state in the final manuscript that these conclusions are speculative and cannot be confirmed with certainty.

- Comment 28: Figure 16: it is quite difficult and even speculative to claim the presence of the LSB on the lower Re numbers by looking at these figures due to these discrepancies in the pressure readings. It could help if you add static polars for the same cases. **Response:** This comment is closely related to comment 27 and will be addressed accordingly.
- Comment 29: Line 349 "As this increases"... what is referred here?

Response: We refer to the Reynolds number. We amended the revised manuscript accordingly.

• Comment 30: Line 352 how do the polars show clear suction peak? Polars only show loads.

**Response:** We meant "the negative values of the pressure distributions" rather than the polars. This was an error and was corrected in the revised manuscript.

• Comment 31: Line 355 where is the detachment and reattachment?

**Response:** The authors are not sure about this comment's meaning. Is the reviewer asking for x/c coordinates at which, for different k and Re, the boundary layer detaches and reattaches?

• Comment 32: Line 359 if the suction peak magnitude is responsible for the lift increase which is correct, which case is referred to in line 355 where trailing edge provides the lift.

**Response:** The dynamic stall mechanism is a combination of higher suction peak magnitude and, for low angles of attack, rapid trailing edge stall. In line 355, we identify the trailing edge reattachment as the responsible for the increase in transient nose-down forces.

• Comment 33: The drag and moment data and results from zigzag tape are missing.

**Response:** Comment addressed in the initial discussion section of this document. These data are not included in the manuscript. The text was adjusted accordingly.